# International professional practices in mental health, organization of psychiatric care, and COVID-19: A survey protocol

Laurence Fond-Harmant[1], Hélène Kane[2], Jade Gourret Baumgart[2], Emmanuel Rusch[2], Hervé Breton[2], Wissam el-Hage [3], Jocelyn Deloyer[4], Marie-Clotide Lebas[5], Donatella Marazziti[6], Johannes Thome[7], Frédéric Denis[2]*

1 Agence de Coopération Scientifique Europe-Afrique-Luxembourg and LEPS, Laboratoire Education et Pratiques en Santé, Université Sorbonne Paris Nord, Paris, France, 2 Faculté de Médecine, EA 75–05 Education, Ethique, Santé, Université François-Rabelais, Tours, France, 3 CIC 1415, INSERM, Centre d'Investigation Clinique, CHRU de Tours, Tours, France, 4 Centre Neuro Psychiatrique St. Martin, Dave Namur, Belgium, 5 Département des Sciences de la Santé Publique et de la Motricité, Haute Ecole de la Province de Namur, Namur, Belgium, 6 Department of Experimental and Clinical Medicine, Section of Psychiatry, University of Pisa, Unicamillus University of Rome and Brain Research Foundation, Lucca, Italy, 7 Department of Psychiatry, University of Rostock, Rostock, Germany

* frederic.denis@chu-tours.fr

**Funding:** This trial is funded by the National Research Agency (Agence National de la Recherche) and the Region Centre Val de Loire,

## Abstract

### Objective

Our project aims to provide:

1. an overview of the impact of the COVID-19 pandemic on the field of mental health professionals in 23 countries;

2. a model of recommendations for good practice and proposals for methods and digital tools to improve the well-being at work of mental health professionals and the quality of services offered during crisis and post-crisis periods;

3. an in-depth ethics review of the assessment of the use of numerical tools for psychiatry professionals and patient support, including teleconsulting.

### Methods

This is a large international survey conducted among 2,000 mental health professionals in 23 countries over a 12-month period. This survey will be based on 30 individual interviews and 20 focus group sessions, and a digital questionnaire will be sent online to 2,000 professionals based on the criteria of gender, age, professional experience, psychiatric specialty, context of work in psychiatry, and geographical location. Regarding the development of telepsychiatry during the COVID-19 pandemic, a pilot study on the use of digital tools will be carried out on 100 clients of psychiatry professionals in France and Belgium.

### Discussion-conclusion

This study will contribute to the co-construction of an international organization and monitoring system that takes into account psychiatric health professionals as major resources to

France). The funders had and will not have a role in study design, data collection and analysis, decision to publish, or preparation of the manuscript.

**Competing interests:** The authors have declared that no competing interests exist.

fight against the COVID-19 pandemic and to develop efficient processes for preparing and anticipating crises by reducing psychosocial risks as much as possible. This project also aims to design tools for remote medicine and to develop the use of numerical tools for monitoring and supporting professionals and helping professionals to build the conditions for satisfactory operational work during crises and post-crisis situations, using adapted organizational methods. Our ongoing research should support professionals in the search for existing concrete solutions to cope with emergency work situations while maintaining an optimal quality of life.

## Introduction

The first case of SARS-CoV-2 infection was identified in China on November 17, 2019; Europe was the epicenter for a period in February and March 2020; and the WHO declared a pandemic on March 11, 2020 [1]. In the context of work overload, health care professionals are faced with a situation of helping and caring for others while being exposed to the disease themselves [2]. To draw a parallel with SARS-CoV-2, studies of the 2003 SARS outbreak revealed that health care professionals who had friends or relatives infected with SARS were two to three times more likely to experience elevated post-traumatic symptoms than those who had not been directly exposed to the infection themselves. During this period of SARS, health professionals expressed deep concern for their loved ones, colleagues, and themselves in terms of their safety in the health care field. They exhibited signs of anger, anxiety, and stress related to the risk of contamination when providing care and the uncertainty generated by the situation in general [3]. Lasting negative effects were observed several years after SARS, such as alcohol-related symptoms, insomnia, and depressive disorders [3–5].

This unprecedented health crisis has raised questions about the role of psychiatry in assisting the population in dealing with the stress generated by the uncertainty and loss of reference points, in particular during lockdown and post-lockdown periods. Psychiatric-care professionals find themselves in the dual position of having to defend the urgent and essential needs of their patients and also having to prove their resistance and resilience in the social or family ordeal they are experiencing and imposing on their relatives. Nevertheless, psychiatric-care professionals can provide significant contributions in overcoming the pandemic by supporting other medical disciplines and absorbing the psychological impact of the event.

These successive events have had direct and indirect impacts on inpatient and outpatient organizations. In order to deal as effectively as possible with these stress and trauma management challenges, the scientific literature [3–6] recommends a number of interventions, notably:

a. Clarity in the communication of guidelines and precautionary measures;

b. Capacity to dialogue with establishment management in order to obtain its support;

c. Support from supervisors and co-workers;

d. Family support;

e. Capacity to talk to someone about their experiences;

f. Religious beliefs [7].

In order to obtain a real-time overview of the psychiatry and mental-health service status in Europe, Israel, and Turkey at the beginning of March 2020, we set up an ad hoc group of 23 contributors willing to participate in the review by describing the situation in their countries of origin by means of interviews and questionnaire feedback.

The preliminary results of the pre-study carried out from March to May 2020 identified several challenges facing mental health professionals and mental health services during the COVID-19 pandemic. These include [8,9]:

1. Access to psychiatric diagnosis and treatment should be maintained despite increasingly difficult circumstances.

2. Mental health services must be provided to patients who test positive for COVID-19. Various treatment regimens need to be developed for patients with and without respiratory symptoms with clear rules when a psychiatric patient needs to be transferred to respiratory care services. The question must be addressed of how to treat patients who need to remain in psychiatric clinics (isolation required, employee protection, possible need to establish a specific psychiatric service for COVID-19-positive patients).

3. For psychiatric patients treated in respiratory or intensive care units, functioning psychiatric liaison services should be provided and possibly increased; meanwhile, routine services for patients with somatic and psychiatric symptoms should be maintained.

4. It is also important to take measures to protect particularly vulnerable psychiatric patients, such as those treated in geriatric psychiatry departments, because the elderly are much more likely to develop severe symptoms when they are infected with COVID-19. As evidence from China indicates [10], there is a considerable risk that insufficient and inadequate attention will be given to this specific and vulnerable group of patients.

5. Psychiatric and mental health personnel need to be informed and trained in the control of infectious diseases, an area in which they generally have limited expertise. Innovation projects, for example, in the form of mentoring, are necessary and should be modeled in Europe because economic or structural crises affect training content, entry into employment, and qualifications [11].

6. With regard to the risk of infection of psychiatric and mental health professionals, the increased use of electronic tools that enable telepsychiatry has the potential to play a major role. These tools should be introduced in an ethical manner, both for the psychiatric professionals and users as well as for research in this field.

7. Both staff and patients must be frequently informed and reminded of the basic hygiene measures that are essential to reduce the spread of infectious diseases.

8. Psychiatric hospitals also have a duty to provide services to health care professionals involved in the fight against COVID-19 who develop secondary psychiatric symptoms, such as stress-related disorders or signs of anxiety or panic [12].

9. There is a significant risk that the need for psychiatric services will increase considerably at a time when services and their staffs are under intense stress. The current form of the organization of service provision must be restructured to meet the needs of this specific emergency situation, in compliance with ethical and therapeutic frameworks.

10. Psychiatric and mental health professionals need to liaise with decision-makers in order to ensure that the special needs of patients with mental and psychological disorders are taken into account when emergency measures are implemented.

11. In addition to online psychological consultation services, there are indications that digital psychiatry also allows for rigorous online monitoring of psychological health and mental health education [13]. This kind of research is essential in order to learn from the current crisis and for the future by integrating the ethical dimensions and those of the fight against social inequalities in health. For many of these challenges, specific solutions need to be developed at European, national, and regional levels. To address the challenges now and in the future, the solutions must take into account the structure of specific services in each area (e.g., availability or lack of inpatient facilities, role of community psychiatry, etc.).

In this context, our priority now is to complete our preliminary work to better understand the psychological and societal impacts of the COVID-19 health crisis on psychiatry professionals (PP), their professional positions, and their work organizations so as to provide the keys to understanding and the means to respond to crisis and post-crisis situations effectively and operationally in the future, based on the experience and the adjustment of an ethical professional position.

Our project, Psychiatry Professionals and COVID-19 in Europe: Psychological Impact Management and Crisis and Post-Crisis Organization (Psy-GIPOC), aims to provide:

1. an overview of the impact of the COVID-19 pandemic on the field of mental health professionals in 24 countries;

2. a model of recommendations for good practice and proposals for methods and digital tools to improve the well-being at work of mental health professionals and the quality of services offered during crisis and post-crisis periods;

3. an in-depth ethics review of the assessment of the use of numerical tools for psychiatry professionals and patient support, including teleconsulting.

## Study design and methods

### Study design

This is a large international survey conducted among 2,000 professionals in 23 countries over a 12-month period to draw up an assessment of the experience of mental health professionals in order to identify their needs in terms of methods and tools that facilitate their work in crisis and post-crisis periods. This mixed-method survey is qualitative and quantitative: 30 semi-structured individual interviews and 20 focus group sessions plus a digital questionnaire to be sent online to 2,000 psychiatry and mental health professionals, based on criteria of gender, age, professional experience, psychiatric specialties, context of work in psychiatry, and geographic location. Simultaneously, a pilot study on the use of digital tools will be carried out on clients of 100 psychiatry professionals in France and Belgium.

### Creation of a scientific committee

The coordination of the research will be under the control of a scientific committee composed of specialist researchers (public health workers, psychiatrists, psychologists, mental health specialists, ethics and health organizations, pertinent engineering professionals) with expertise in issues related to improving professional practices in psychiatry in a variety of contexts. The scientific committee is made up of Belgian, French, German, Italian, and Luxembourgish academics and staff at three psychiatric hospitals. All have higher academic degrees (six MDs, eight PhDs). Of the respondents, three were psychiatrists, one was a specialist in psychiatric nursing, one was a psychologist, four were allied health professionals working in mental health, and four were public health researchers specializing in psychiatry and mental health.

### First step: Questionnaire grids creation

At the beginning of the study, exploratory interviews will be carried out with the psychiatry professionals on the steering committee. These exploratory interviews will provide opportunities for descriptions of behaviors, situations, and emotions in real-life situations. At the same time, a broad review of the international literature will be undertaken on the following themes: COVID-19; mental health services organization; organization of psychiatric care; psychiatric and mental health professionals; e-health; mental health; ethics. A cross-referencing of the most salient themes will serve as a basis for the creation of three grids for a questionnaire for the online survey of 2,000 people in 23 countries:

1. An evaluation grid of digital tools in mental health;

2. A second grid for 30 semi-structured individual interviews with professionals;

3. A third grid for 20 focus groups of mental health professionals [14].

    The questionnaire (Additional file 1) aims to investigate the following themes:

- Local organizational adaptations during the pandemic for the continuation of care in compliance with health and ethical conditions;

- The modalities of use of telepsychiatry and online mental health monitoring and the impact of the use of these tools in interprofessional relations and in the care relationship;

- The impact of COVID-19 on working conditions in psychiatry and on the mental health of professionals in this sector.

### Second step: A feasibility study

A feasibility study with 50 professionals in France and 50 in Belgium will be carried out to test the online questionnaire that will be sent to 2,000 professionals as well as the grids for the 30 semi-structured individual interviews and the 20 focus groups [14]. The elements evaluated will be the time taken to complete the questionnaire and the level of understanding of the questions by the participants.

    This validation stage will also identify the existing digital tools used by the professionals and the ethical frameworks in which they are used.

### Third step: International online survey

This survey will be conducted online among 2,000 professionals in 23 countries (Belgium, Spain, Finland, France, Germany, Greece, Iceland, Ireland, Israel, Italy, Luxembourg, Malta, Netherlands, Poland, Portugal, Czech Republic, Romania, Russia, Sweden, Switzerland, Turkey, Ukraine, United Kingdom). The results of this quantitative study will be amplified with the 30 semi-structured individual interviews and 20 focus groups. The flow chart of the study is shown in Fig 1.

## Methods

### Selection criteria for the questionnaires among countries

A roster of 2,000 health professionals in psychiatry (HPP), matched according to age, gender, experience, and geographic location, will be compiled using snowball sampling from the networks of ad hoc professionals and researchers mobilized in March 2020 [11]. Thus, 23 experts will provide a database of HPP to whom to send the questionnaire or will relay the

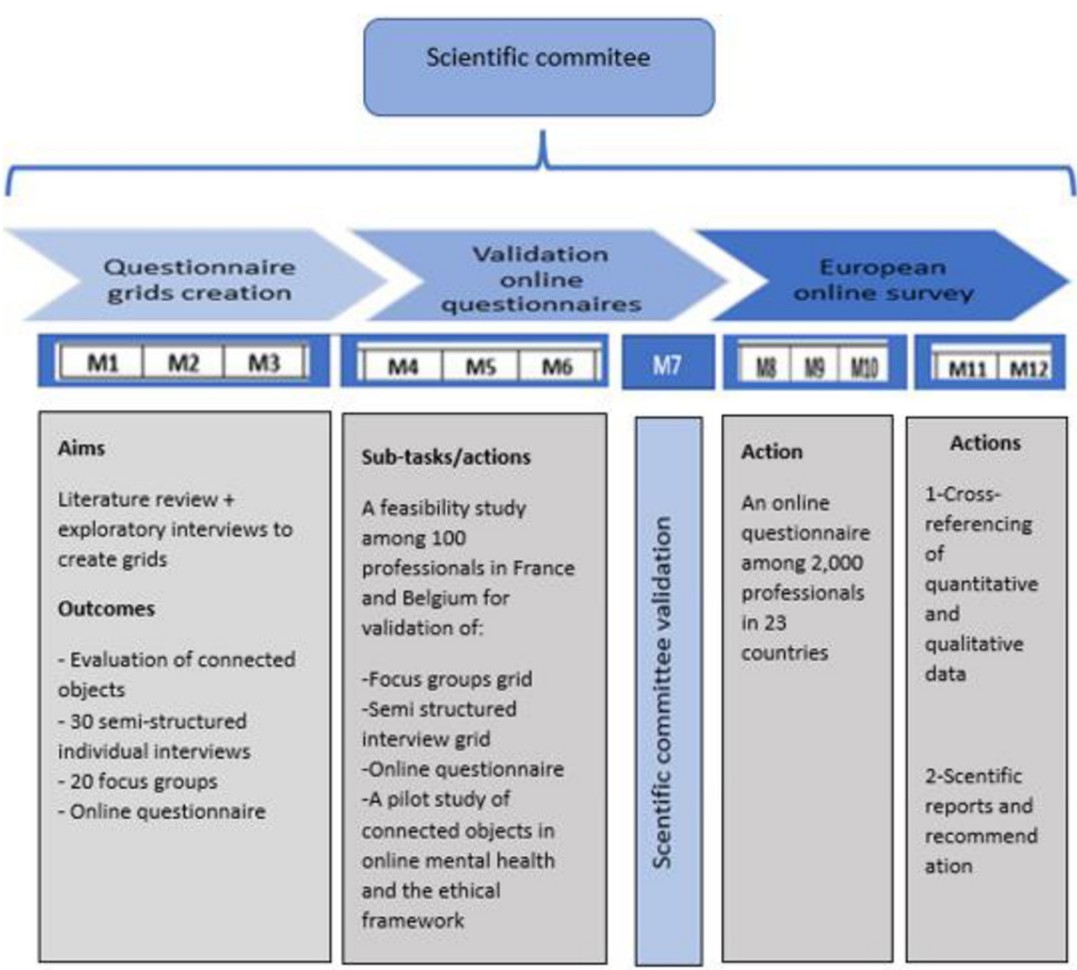

**Fig 1. Flow chart of Psy-GIPOC study.**

questionnaire directly to the professionals, according to the target number of respondents per country, i.e., a total of approximately 2,200 professionals.

After discussion with the scientific committee, we decided that the number of professionals participating in the study would be based on the population threshold of the involved countries with an overestimation of 10% after considering the number of psychiatric hospital beds per country [15], the number of psychiatrists per 100,000 inhabitants [16], and the heterogeneity of this data (Table 1).

## Distribution of the questionnaires among countries

The distribution of the questionnaires among countries has at least three objectives:

1. To allow for a diversity of respondents in each country among medical professionals, para-medical professionals, social work professionals, and others such as psychologists;

2. To maximize the chance of obtaining a total of 2,000 respondents;

3. To ensure that the sample is representative of the geographic area concerned, even if this objective is difficult to achieve because of the non-standardization of occupational

**Table 1. Selection criteria for the questionnaires among countries.**

| Country | Population in millions of inhabitants | Psychiatric hospital beds | Psychiatrists per 100,000 inhabitants | The population threshold | Number of HPP* selected by country |
|---|---|---|---|---|---|
| Iceland | 0.364134 | 36.57 | 23.53 | <10 | 60 |
| Malta | 0.514564 | 105.03 | 11.35 | <10 | 60 |
| Luxembourg | 0.626108 | 74.18 | 21.46 | <10 | 60 |
| Ireland | 4.96 | 33.57 | 16.56 | <10 | 60 |
| Finland | 5.53 | 53.67 | - | <10 | 60 |
| Switzerland | 8.6 | 93.07 | 52.31 | <10 | 60 |
| Israel | 8.88 | - | - | <10 | 60 |
| Sweden | 10.33 | 41.25 | 23.46 | <20 | 90 |
| Portugal | 10.29 | 63.6 | 13.41 | <20 | 90 |
| Czech Republic | 10.69 | 92.75 | 14.79 | <20 | 90 |
| Greece | 10.71 | 73.92 | 25.79 | <20 | 90 |
| Belgium | 11.55 | 135.22 | 17.23 | <20 | 90 |
| Netherlands | 17.41 | 85.58 | 24.15 | <20 | 90 |
| Romania | 19.32 | 85.44 | 11.88 | <20 | 90 |
| Poland | 37.96 | 62.23 | 9.23 | <100 | 120 |
| Ukraine | 41.98 | - | - | <100 | 120 |
| Spain | 47.33 | 36.1 | 10.93 | <100 | 120 |
| Italy | 60.24 | 8.87 | 17.08 | <100 | 120 |
| United Kingdom | 67.03 | 36.9 | 17.98 | <100 | 120 |
| France | 67.01 | 82.69 | 22.87 | <100 | 120 |
| Turkey | 82 | 4.77 | 5.45 | <100 | 120 |
| Germany | 83.15 | 128.45 | 27.45 | <100 | 120 |
| Russia | 144.5 | - | - | >100 | 190 |
| Total of professionals to be included in the study with an overestimation of 10% | | | | | **2,200** |

*HPP: Health psychiatry professionals.

categories among countries and the lack of common indicators among countries inside and outside Europe.

All participants should work in the field of mental health/psychiatry (e.g., as a psychiatrist, psychologist, researcher, manager) or have sufficient insight into these services (e.g., as an academic collaborating with psychiatric service clinicians). They will have a comprehensive knowledge of the situation in their country during the COVID-19 crisis (i.e., beyond the situation of their institution), and they will be prepared to fill out a questionnaire within three days. The objective of standardizing occupational categories across countries will be difficult to achieve, for our list of study participants, in the absence of common indicators across countries inside and outside Europe. The initial version of the questionnaire will be designed in French; this version will be distributed in some countries, including France. The questionnaire has been translated into the official language(s) of the different countries involved in the study: English, German, Portuguese, Spanish, Finnish, Greek, Hebrew, Hindi, Italian, Dutch, Polish, Romanian, Russian and Ukrainian. The questionnaire is also available to all respondents in English.

SPHINX© software will be used to mobilize respondents and disseminate the survey to process and analyze data.

## Data management of interview and focus group

The analysis of the interviews will proceed in six distinct main stages, summarized here:

1. Open codification of re-transcribed interviews in order to identify as many topics as possible from the initial corpus;

2. Categorization of the codified elements: careful reading of the entire corpus so that each category is clearly defined, its properties revealed, and the different forms and conditions of occurrence of the specified phenomena identified;

3. Linking categories: writing more detailed memos and designing explanatory diagrams;

4. Integration of the previous steps in order to identify the essence of the phenomenon;

5. Modeling the phenomenon: in addition to being described, defined, and explained, its dynamics will be examined and conceptualized, after which the structural and functional relationships of each of its constituents will be highlighted;

6. Theorization: a thorough and exhaustive construction will be undertaken of the "multi-dimension" and "multi-causality" of the phenomenon of associations among the needs, expectations, and representations of the different groups (health care users, health care fields, primary care and mental health professionals).

Qualitative data management will be done using NVIVO$^{©}$ software.

## Cross-referencing of quantitative and qualitative data

Interdisciplinary meetings with the steering committee will be organized throughout the analysis process; during these meetings, the framework will be adjusted according to the standard indications of the grounded theory, if necessary. Triangulation of the data by researchers from different fields as well as experts will guarantee a high level of both internal and external validity of the results once freed from the theoretical paradigms. To put it another way, this is not only a study *about* but also a study *with* these professionals.

## Ethical considerations and dissemination

This research received ethical approval from the ethics committee of the University Hospital of Tours on February 4, 2021, and is registered under number 2020 006.

The project outcomes will be disseminated through selected peer-reviewed journals, conference presentations, workshops, and webinars.

## Discussion-conclusion

All recent work on major epidemics links the concepts of territory, epidemiology, and the health care system. There is a geography of virus development. These concepts show the links among the progression of the virus, the organization of care, and the state of the health care system. They show the territorial disparities and health factors that contribute to inequalities in access to health care systems and the vulnerability of fragile populations. This situation has destroyed the relationship of trust in hospitals and clinics: it is estimated that 50% of deaths are not due, for example, to Ebola but to other non-treated diseases [13,17].

The quality of psychiatric services during the COVID-19 pandemic varies from country to country, and public health policies and professional contexts differ. This research will shed light on this geopolitical dimension of COVID-19. The project will provide an intelligent

international cross-look that will allow the exchange of innovative practices in a very operational way.

This study will contribute to the co-construction of an international organization and monitoring system that takes into account psychiatric health professionals as major resources to fight against COVID-19 and future pandemics [18]. In a context where the effects of such health crises have a direct impact on the development of the needs for psychiatric services during the crises and several years afterward [19], it is critical to develop research work that sheds light on these two interrelated dimensions.

The objective is to map out a high-quality level of existing structures and the availability of emergency structures with human resources in good physical and mental health. The aim is to draw lessons in order to develop efficient processes for preparing for and anticipating crises by reducing psychosocial risks as much as possible. This project also aims to design tools for remote medicine and to develop the use of digital tools for monitoring and supporting professionals. The aim of these digital tools will be to prevent psychological suffering and its consequences [20,21] while accompanying identifying signs of ill-being. The goal is to help professionals build the conditions for satisfactory operational work during crises and post-crisis situations, using adapted organizational methods. Our ongoing research should support professionals in the search for existing concrete solutions to cope with emergency work situations while maintaining an optimal quality of life.

## Supporting information

**S1 File. Interview grid + Questionnaire.**
(PDF)

## Acknowledgments

The authors are grateful to the National Research Agency (Agence National de la Recherche) and the Region Centre-Val de Loire. The authors would like to especially thank the psychiatric health professionals for their contributions.

## Author Contributions

**Conceptualization:** Laurence Fond-Harmant, Frédéric Denis.

**Data curation:** Hélène Kane, Jade Gourret Baumgart.

**Supervision:** Jocelyn Deloyer.

**Validation:** Emmanuel Rusch, Hervé Breton, Marie-Clotide Lebas, Donatella Marazziti, Johannes Thome.

**Visualization:** Wissam el-Hage.

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
