## [Decision Letter · Decision Letter 0]

24 Aug 2021

PONE-D-21-09107

International professional practices in mental health, organization of psychiatric care, and Covid-19: A survey protocol

PLOS ONE

Dear Dr. Denis,

Thank you for submitting your manuscript to PLOS ONE. After careful consideration, we feel that it has merit but does not fully meet PLOS ONE’s publication criteria as it currently stands. Therefore, we invite you to submit a revised version of the manuscript that addresses the points raised during the review process.

We look forward to receiving your revised manuscript.

Kind regards,

Vincenzo De Luca

Academic Editor

PLOS ONE

Journal Requirements:

5. Your abstract cannot contain citations. Please only include citations in the body text of the manuscript, and ensure that they remain in ascending numerical order on first mention.

6. We note that Figure 2 in your submission contain map images which may be copyrighted. All PLOS content is published under the Creative Commons Attribution License (CC BY 4.0), which means that the manuscript, images, and Supporting Information files will be freely available online, and any third party is permitted to access, download, copy, distribute, and use these materials in any way, even commercially, with proper attribution. For these reasons, we cannot publish previously copyrighted maps or satellite images created using proprietary data, such as Google software (Google Maps, Street View, and Earth). For more information, see our copyright guidelines: http://journals.plos.org/plosone/s/licenses-and-copyright.

Reviewers' comments:

Reviewer's Responses to Questions

**Comments to the Author**

1. Does the manuscript provide a valid rationale for the proposed study, with clearly identified and justified research questions?

Reviewer #1: Yes

2. Is the protocol technically sound and planned in a manner that will lead to a meaningful outcome and allow testing the stated hypotheses?

Reviewer #1: Partly

3. Is the methodology feasible and described in sufficient detail to allow the work to be replicable?

Reviewer #1: Yes

4. Have the authors described where all data underlying the findings will be made available when the study is complete?

Reviewer #1: Yes

5. Is the manuscript presented in an intelligible fashion and written in standard English?

Reviewer #1: No

6. Review Comments to the Author

You may also provide optional suggestions and comments to authors that they might find helpful in planning their study.

Reviewer #1: PlosOne Manuscript Number PONE-D-21-09107

Title: “International professional practices in mental health organization of psychiatric care, and Covid-19: A survey protocol

Overview:

This study protocol aims to investigate the impact of the Covid-19 pandemic on mental health care providers/professionals using survey data from 2,000 individuals located across 23 countries. The data collected will be used to develop tools to improve mental health care delivery during a global health crisis, as well as specific monitoring and support tools for mental health professionals. Overall, the proposed work addresses an important topic, as the world looks to recover from this pandemic and prepare for future public health crises. The writing and presentation, however, make it very unclear what has already been done, versus what aspects of this study have yet to be completed.

Specific comments:

Introduction-

1. In my opinion, too much time is spent discussing the impact of COVID-19 pandemic on the health care community in general, as this has been well described in the academic literature as well as the popular media. The focus here is more specifically mental health care providers and organizations.

Methods-

1. Since a standard questionnaire was not used, access to the full final survey should be provided as a supplement, or at least specific categories and examples of questions from each category. I see that there is an included supplemental material file, but it is not referenced in the body of the manuscript, so it is rather unclear what this table represents.

2. Figure 1- What happened to M7? Is there a more specific timeline (April 2021-March 2022, for example)?

3. Figure 2- this figure is not necessary; add a column to Table 1 that includes the actual number of HPP included in the

survey.

4. Given the international nature of this study, what language will/has been used for data collection and data management?

7. PLOS authors have the option to publish the peer review history of their article (what does this mean?). If published, this will include your full peer review and any attached files.

Reviewer #1: No

---

## [Author Response · Author response to Decision Letter 0]

31 Aug 2021

Response to Reviewers

Reviewer #1: PlosOne Manuscript Number PONE-D-21-09107

Title: “International professional practices in mental health organization of psychiatric care, and Covid-19: A survey protocol

Overview:

This study protocol aims to investigate the impact of the Covid-19 pandemic on mental health care providers/professionals using survey data from 2,000 individuals located across 23 countries. The data collected will be used to develop tools to improve mental health care delivery during a global health crisis, as well as specific monitoring and support tools for mental health professionals. Overall, the proposed work addresses an important topic, as the world looks to recover from this pandemic and prepare for future public health crises. The writing and presentation, however, make it very unclear what has already been done, versus what aspects of this study have yet to be completed.

Specific comments:

Introduction-

1. In my opinion, too much time is spent discussing the impact of COVID-19 pandemic on the health care community in general, as this has been well described in the academic literature as well as the popular media. The focus here is more specifically mental health care providers and organizations.

Response:

As requested, we have revised the "introduction" section to focus more on mental health care providers and organizations. We also revised the “references” section.

Methods-

1. Since a standard questionnaire was not used, access to the full final survey should be provided as a supplement, or at least specific categories and examples of questions from each category. I see that there is an included supplemental material file, but it is not referenced in the body of the manuscript, so it is rather unclear what this table represents.

Response:

The study questionnaire (Additional file 1) aims to investigate the following themes:

- Local organizational adaptations during the pandemic for the continuation of care in compliance with health and ethical conditions; 

- The modalities of use of telepsychiatry and online mental health monitoring and the impact of the use of these tools in interprofessional relations and in the care relationship; 

- The impact of COVID-19 on working conditions in psychiatry and on the mental health of professionals in this sector. We have added this paragraph to line 143.

2. Figure 1- What happened to M7? Is there a more specific timeline (April 2021-March 2022, for example)?

Response:

Thank you for pointing this mistake. M7 corresponds to the time spent validating the survey tools with the scientific committee. We have modified Figure 1 to clarify this point.

3. Figure 2- this figure is not necessary; add a column to Table 1 that includes the actual number of HPP included in the survey.

Response:

We agree that the information given in Figure 2 is redundant with that in Table 1 in the column entitled: Number of selected HPPs* per country. As suggested, we have removed Figure 1.

4. Given the international nature of this study, what language will/has been used for data collection and data management?

Response:

The initial version of the questionnaire will be designed in French, and this version will be distributed in some countries including France. The questionnaire has been translated into the official language(s) of the different countries involved in the study: English, German, Portuguese, Spanish, Finnish, Greek, Hebrew, Hindi, Italian, Dutch, Polish, Romanian, Russian and Ukrainian. The questionnaire is also available to all respondents in English.We have added this sentence Page 9, line 190.

We thank the reviewers for giving us an opportunity to substantially improve the content and the presentation of our manuscript.

---

## [Decision Letter · Decision Letter 1]

23 Nov 2021

PONE-D-21-09107R1International professional practices in mental health, organization of psychiatric care, and Covid-19: A survey protocolPLOS ONE

Dear Dr. Denis,

Thank you for submitting your manuscript to PLOS ONE. After careful consideration, we feel that it has merit but does not fully meet PLOS ONE’s publication criteria as it currently stands. Therefore, we invite you to submit a revised version of the manuscript that addresses the points raised during the review process.

We look forward to receiving your revised manuscript.

Kind regards,

Sanjay Kumar Singh Patel, Ph.D.

Academic Editor

PLOS ONE

Journal Requirements:

Reviewers' comments:

Reviewer's Responses to Questions

**Comments to the Author**

1. Does the manuscript provide a valid rationale for the proposed study, with clearly identified and justified research questions?

Reviewer #1: Yes

Reviewer #2: Yes

2. Is the protocol technically sound and planned in a manner that will lead to a meaningful outcome and allow testing the stated hypotheses?

Reviewer #1: Yes

Reviewer #2: Yes

3. Is the methodology feasible and described in sufficient detail to allow the work to be replicable?

Reviewer #1: Yes

Reviewer #2: Yes

4. Have the authors described where all data underlying the findings will be made available when the study is complete?

Reviewer #1: No

Reviewer #2: Yes

5. Is the manuscript presented in an intelligible fashion and written in standard English?

Reviewer #1: No

Reviewer #2: Yes

6. Review Comments to the Author

You may also provide optional suggestions and comments to authors that they might find helpful in planning their study.

Editor comments:

The manuscript requires English Proof.

Reviewer #1: Summary:

This work aims to build on preliminary work to better understand the psychological and societal impacts of the Covid-19 pandemic on psychiatry professionals and mental health services/organizations in order to inform best practices regarding psychiatric support and recovery during this, and future public health crises. While the study and proposed work address important questions, the writing lacks clarity and is full of typos, making it difficult for this reviewer to fully assess the scientific content of this manuscript.

Specific comments:

Introduction

1. The final sentence of the first paragraph (lines 9-12) is unclear (and appears to have a missing word, or typo?). Furthermore, how can the effects have already lasted “several years after the trauma” when the pandemic was declared less than 2 years ago:

They exhibited signs of anger, anxiety, and stress related to the risk of contamination and the uncertainty in daily routines [3] and lasting negative effects several years after the trauma like include symptoms related to alcohol consumption, insomnia, and depression [3-5].

Study Design and Methods

2. Please provide clarification regarding the difference between “questionnaire grids” versus “themes” in the text. For example, are each of the themes being examined in each grid?

3. Please explain what is meant by “20 focus groups of mental health professionals.”

4. The terms inside parentheses on lines 137-138: What is this in referencing? For example, is this initial phase meant to address the feasibility of delivering the questionnaire, in terms of duration/time to deliver and complete, clarity of the questions, etc.?

5. Figure 1: Please do not use acronyms/abbreviations in the figure title, especially if they have not been introduced in the text. Please check flow chart text for typos and standardize the formatting/organization of the text; perhaps consider dividing each box into explicit "actions", "sub-tasks/actions" and "outcomes/aims"

Reviewer #2: Authors of article entitled "International professional practices in mental health, organization of psychiatric care, and Covid-19: A survey protocol" have addressed suggested comments.

---

## [Author Response · Author response to Decision Letter 1]

29 Nov 2021

Editor comments:

The manuscript requires English Proof.

Response: 

The English version of this manuscript was corrected by the “American Manuscript Editors”. An "English Editing Certificate" was associated with the submission of this revised version.

Reviewer #1: Summary:

This work aims to build on preliminary work to better understand the psychological and societal impacts of the Covid-19 pandemic on psychiatry professionals and mental health services/organizations in order to inform best practices regarding psychiatric support and recovery during this, and future public health crises. While the study and proposed work address important questions, the writing lacks clarity and is full of typos, making it difficult for this reviewer to fully assess the scientific content of this manuscript.

Response: 

We thank the reviewer for giving us an opportunity to substantially improve the content and the presentation of our manuscript with constructive comments.

Introduction

1. The final sentence of the first paragraph (lines 9-12) is unclear (and appears to have a missing word, or typo?). Furthermore, how can the effects have already lasted “several years after the trauma” when the pandemic was declared less than 2 years ago:

They exhibited signs of anger, anxiety, and stress related to the risk of contamination and the uncertainty in daily routines [3] and lasting negative effects several years after the trauma like include symptoms related to alcohol consumption, insomnia, and depression [3-5].

Response:

To introduce this paragraph, we have given some general information on Sars-CoV2. Without precise knowledge of the consequences of this infection, we have compared it to the SARS epidemic in order to highlight certain elements already known about the working conditions of health professionals in such a context.

We have made some corrections using the track changes, lines 2 to 12. We hope this paragraph will be clearer

Study Design and Methods

2. Please provide clarification regarding the difference between “questionnaire grids” versus “themes” in the text. For example, are each of the themes being examined in each grid?

Response:

The aim of qualitative research is to develop concepts that help us understand social phenomena in natural (rather than experimental) contexts, focusing on the meanings, experiences and perspectives of all the different participants in the study. The analysis of this information allows themes to emerge that will be further analysed through different techniques such as semi-structured individual interviews and focus groups (with questionnaire grids to explore all the selected themes).

In the “First step: Questionnaire grids creation” section we have added the sentence below.

“These exploratory interviews will allow the description of behaviors, situations and emotions in real life situations.”

Line 125 for clarification we have deleted "results" and replaced it with "themes".

3. Please explain what is meant by “20 focus groups of mental health professionals.”

Response:

This survey protocol (focus groups) makes it possible to collect the opinions of several people. This technique also makes it possible to study the social relations between the people present. A number of 20 persons is needed for data saturation.

We have added a reference to justify this point lines 133 and 146.

[14]. Marshall B, Cardon P, Poddar A, Fontenot R. Does Sample Size Matter in Qualitative Research? A Review of Qualitative Interviews in this Research. Journal of computer information systems.2013; 54(1), 11-22.

And listed it in the reference section.

4. The terms inside parentheses on lines 137-138: What is this in referencing? For example, is this initial phase meant to address the feasibility of delivering the questionnaire, in terms of duration/time to deliver and complete, clarity of the questions, etc.?

Response:

To clarified this point, we deleted the sentence in parentheses and added line 146 “The elements evaluated will be the time taken to complete the questionnaire and the good understanding of the questions by the participants.” 

5. Figure 1: Please do not use acronyms/abbreviations in the figure title, especially if they have not been introduced in the text. Please check flow chart text for typos and standardize the formatting/organization of the text; perhaps consider dividing each box into explicit "actions", "sub-tasks/actions" and "outcomes/aims"

Response:

Thanks for the comment. Line 89, we introduced the acronym of our study: Psy-GIPOC. 

As requested, we checked flow chart text for typos and standardize the formatting/organization of the text. We thank the reviewer for suggesting that each box be divided into explicit "actions", "subtasks/actions" and "outcomes/objectives".

Reviewer #2: Authors of article entitled "International professional practices in mental health, organization of psychiatric care, and Covid-19: A survey protocol" have addressed suggested comments.

Response:

Thank you

---

## [Editor Report · Decision Letter 2]

13 Dec 2021

International professional practices in mental health, organization of psychiatric care, and COVID-19: A survey protocol

PONE-D-21-09107R2

Dear Dr. Denis,

We’re pleased to inform you that your manuscript has been judged scientifically suitable for publication and will be formally accepted for publication once it meets all outstanding technical requirements.

Kind regards,

Sanjay Kumar Singh Patel, Ph.D.

Academic Editor

PLOS ONE
---

## [Editor Report · Acceptance letter]

16 Dec 2021

PONE-D-21-09107R2 

International professional practices in mental health, organization of psychiatric care, and COVID-19: A survey protocol 

Dear Dr. Denis:

I'm pleased to inform you that your manuscript has been deemed suitable for publication in PLOS ONE. Congratulations! Your manuscript is now with our production department. 

Kind regards, 

on behalf of

Dr. Sanjay Kumar Singh Patel 

Academic Editor

PLOS ONE